# Effect of the 16S rRNA Gene Hypervariable Region on the Microbiome Taxonomic Profile and Diversity in the Endangered Fish *Totoaba macdonaldi*

**DOI:** 10.3390/microorganisms12112119

**Published:** 2024-10-23

**Authors:** Itzel Soledad Pérez-Bustamante, Roberto Cruz-Flores, Jesús Antonio López-Carvallo, Samuel Sánchez-Serrano

**Affiliations:** 1Centro de Investigación Científica y de Educación Superior de Ensenada (CICESE), Carretera Ensenada-Tijuana No. 3918, Zona Playitas, Ensenada 22860, Mexico; itzelpb@cicese.edu.mx; 2Laboratorio de Fisiología y Genética Marina, Departamento de Acuicultura, Facultad de Ciencias del Mar, Universidad Católica del Norte (UCN), Coquimbo 1781421, Chile; jesus.lopez@ucn.cl; 3Facultad de Ciencias Marinas, Universidad Autónoma de Baja California (UABC), Ensenada 22860, Mexico; sanchez.samuel@uabc.edu.mx

**Keywords:** *Totoaba macdonaldi*, microbiome, 16S rRNA, hypervariable regions, amplicon sequencing

## Abstract

Understanding the intricate dynamics of fish microbiota through 16S rRNA amplicon sequencing is pivotal for ecological insights and effective disease management. However, this approach faces challenges including the co-amplification of host mitochondrial sequences and the variability in bacterial composition influenced by the selected 16S rRNA gene regions. To overcome these limitations, we conducted a comprehensive investigation to identify the most suitable 16S rRNA region for bacterial microbial analysis in endangered fish *Totoaba macdonaldi*, an endemic species of significant ecological and economic importance in Mexico. Targeting four distinct hypervariable regions (V1–V2, V2–V3, V3–V4, and V5–V7) of the 16S rRNA gene, we determined the microbial composition within the distal intestine. A total of 40 microbiomes were sequenced. Our findings underscore the critical impact of region selection on the accuracy of microbiota analysis. The V3–V4 region detected the highest number of bacterial taxa and exhibited significantly higher alpha diversity indices, demonstrating the highest taxonomic resolution. This study emphasizes the necessity of meticulous 16S rRNA region selection for fish microbiota analysis, particularly in native species of ecological and economic significance such as the endangered *T. macdonaldi*, where information is limited. Such optimization enhances the reliability and applicability of microbiota studies in fisheries management and conservation efforts.

## 1. Introduction

The 16S ribosomal RNA (rRNA) gene is the most widely used target gene for bacterial identification [1,2,3] and has been employed to identify bacterial populations from various hosts, including algae [4], mollusks [5], crustaceans [6], and fish [7]. This is because it is highly conserved in all prokaryotic cells (bacterial and archaeal) [8], meaning that at least one copy of the gene is present in a genome [9]. In addition, it can be found in every mitochondrial genome, which, over the course of symbiotic evolution, has shed most of its original genes [10,11]. Also, its function (translation of mRNA to proteins) is essential for cellular existence [12] and has not changed over time, which makes it an excellent housekeeping genetic marker [13,14]. The 16S rRNA gene is approximately 1500 base pairs long [15] and has ten highly conserved regions (common among most bacteria) that are separated by nine hypervariable regions (V1–V9) [2,14], which have relatively low levels of conserved sequences [16]. The conserved regions have been effectively utilized to design universal primers, enabling the identification of a broader range of bacterial taxa, including previously unknown bacteria [9,12]. In contrast, the hypervariable regions facilitate the design of specific primers, allowing for the precise identification of particular species [3,14].

Currently, studies of microbial communities are carried out with the amplicon-based next-generation sequencing (NGS) of certain hypervariable regions of the 16S rRNA gene [17,18,19]. The MiSeq Illumina System is among the most widely utilized next-generation sequencing platforms, producing paired-end reads of up to 300 base pairs [20,21]. This results in combined reads of 600 nucleotides, characterized by high accuracy and quality [22]. This approach enables the sequencing of up to three adjacent hypervariable regions of the 16S rRNA gene, utilizing universal primers that bind to conserved regions [21]. Nevertheless, this approach encounters challenges, notably the co-amplification of host mitochondrial sequences, which contain variants of the 16S rRNA gene [10,11]. This co-amplification reduces the specificity and sensitivity of the analysis, particularly when broad-range primers are used and in samples derived from tissues or blood where host DNA is predominant [11,23,24].

Accordingly, Walker et al. [25] demonstrated that a choice of 16S rRNA hypervariable region can significantly affect the proportion of host DNA amplification, obtaining better results with the V1–V2 region. The V3–V4 region is the most used for fish microbiome studies [7,26,27,28], in this sense the microbiome refers to the microbial communities (microbiota) that inhabit in a determined place including their genomes and the surrounding environments [29]. The microbiota associated with the gut of the fish plays a key role in its health, immunity and contributes to its nutritional functions, secreting a wide variety of enzymes that participate in the digestion and absorption of nutrients [30]. Additionally, it is a crucial component of mucosal barrier function, actively competing with and resisting pathogens [31]. It also increases phagocytic activity [32] and enhances the host’s immune system [33].

The totoaba (*Totoaba macdonaldi*) is a vulnerable, carnivorous fish endemic to the Gulf of California and has become a keystone species in this region [34,35]. However, unregulated fishing and overexploitation have led to a dramatic decline in its wild population, resulting in its classification as an endangered species in Mexico (NOM-059-SEMARNAT-2010) [36] and a vulnerable species by the IUCN [37]. In recent years, aquaculture efforts have transformed totoaba cultivation into a significant economic activity, supporting the livelihoods of many fishermen throughout the Baja California peninsula [38]. Today, totoaba cultivation is well established in the aquaculture industry of northwest Mexico, aimed at both enhancing wild stocks and commercial production [39]. However, a major challenge for totoaba aquaculture is developing a diet that meets its protein requirements at an adequate cost without compromising its overall performance [35].

There are limited studies on the bacterial microbiota associated with the gastrointestinal tract of the totoaba. Most research to date has focused on the effects of various diets on the intestinal microbiota. These include studies on the use of different levels of soy protein [27], dietary supplementation with prebiotics and probiotics [40], the implementation of feed additives such as flavonoids and inulin [35], and the impact of the extrusion process on diet [41], among others. Generally, these studies have shown that the bacterial microbiota, identified using the V3–V4 region, is dominated by the phyla Proteobacteria and Firmicutes. Considering that the selection of hypervariable regions of the 16S rRNA gene yields different bacterial taxonomic resolutions [24], which may vary depending on the sample type or ecological niche [42], it is important to choose the appropriate region for analysis. For instance, in samples from distinct ecotopes in Lake Baikal, the V2–V3 region provided higher resolution than the V3–V4 region, particularly for lower-rank taxa such as species and genera [17]. Additionally, a study by Fadeev et al. [42] recommend using the V4–V5 region for amplifying both bacterial and archaeal communities in the Arctic marine environment.

The choice of hypervariable region can significantly affect richness and alpha diversity [43]. For example, in the taxonomic identification of respiratory samples from humans, alpha diversity was significantly higher for the V1–V2, V3–V4, and V5–V7 regions compared to the V7–V9 region [24]. In contrast, the V6–V8 region captured the highest diversity in marine microbial communities compared with the V4–V5 region [44]. These studies illustrate how our understanding of the bacterial microbiota of the ecologically and economically important totoaba might be skewed by the selection of the 16S rRNA regions. To date research on the gut microbiota of totoaba has primarily focused on the utilization of the V3–V4 region [27,35,40,41]. However, no studies have confirmed whether this is the most suitable region for comprehensively and accurately describing the diversity of the bacterial community, which is crucial for ecological insights and effective disease management. To overcome these limitations, we conducted a comprehensive investigation to identify the most suitable 16S rRNA region for bacterial microbial analysis in juveniles of *T. macdonaldi*. In this work, 40 microbiomes were analyzed to compare the effect of four combinations of hypervariable regions (Figure 1) of the 16S rRNA gene, on the taxonomic composition and bacterial diversity.

## 2. Materials and Methods

### 2.1. Sample Collection and DNA Extraction

Five cultured juvenile fish (*Totoaba macdonaldi*) were obtained from the Pisciculture Biotechnology Unit of the Marine Sciences Faculty at the Autonomous University of Baja California, Mexico. The organisms were sacrificed with an overdose of tricaine methanesulfonate anesthetic (100 mg/L) and a puncture in the cephalic region. All procedures strictly adhered to the American Veterinary Medical Association Guidelines for the Euthanasia of Animals. Body weight (57.61 ± 21.45 g) and length measurements (17.32 ± 1.71 cm) were recorded using an electronic balance (Ohaus, Scout Pro, Parsippany, NJ, USA) and an ichthyometer (AQUATIC BIOTECNOLOGY, Puerto de Santa Maria, España), respectively. Each fish’s intestinal tract was carefully excised using sterilized scalpel and tweezers under strictly aseptic conditions. The distal intestine, encompassing a segment of 1.06 ± 0.40 cm along with its digesta content, was meticulously separated and promptly transferred to individual 1.5 mL sterile Eppendorf tubes, each containing 500 µL of 95% molecular biology grade ethanol (Sigma-Aldrich, Burlington, MA, USA). These samples were preserved at 4 °C until subsequent analysis.

Total DNA was extracted for all samples (*n* = 5) using the FFPE DNA Purification Kit (NORGEN BIOTEK CORP, Thorold, ON, Canada) following the manufacturer’s instructions and considering the modifications described by [26]. The deparaffinization step was omitted from the protocol. Furthermore, during the lysate preparation, a doubled amount of proteinase K (20 µL) was utilized instead of the conventional 10 µL, and the incubation period was extended to one hour and a half at a temperature of 55 °C. Subsequently, the concentration and quality of the extracted DNA were assessed using a NanoDrop 2000 spectrophotometer (Thermo Fisher Scientific, Waltham, MA, USA).

### 2.2. 16S rRNA Gene Amplicon Sequencing

The isolated DNA were sent and processed at OmegaBioservices, located in Norcross, GA, USA. This included primer synthesis, amplification of the 16S rRNA gene, library preparation, and sequencing, adhering strictly to the established protocol for 16S Metagenomic Sequencing Library Preparation (Illumina, Inc., San Diego, CA, USA). Each sample was subjected to amplification of four different hypervariable regions utilizing specific primer pairs detailed in Appendix A, with the corresponding PCR conditions provided in Appendix A. Amplification was carried out using the Kapa HiFi PCR kit, and subsequent libraries were prepared. Sequencing was performed on an Illumina^®^ MiSeq platform employing a 2 × 300 bp paired-end protocol. Notably, each region from every fish was sequenced in duplicates to ensure the robustness and reliability of the data obtained.

### 2.3. Bioinformatic and Statistical Analysis

The raw forward and reverse reads were paired, filtered, and merged in Geneious Prime^®^ 2023.2.1 (https://www.geneious.com (accessed on 6 May 2024)) following the Amplicon Metagenomics Tutorial [45]. The sequences were paired using the “set paired reads” option and trimmed with the BBDuk plugin to remove Illumina adapters. Bases with an average quality score below 30 from both ends were trimmed, and all reads shorter than 10 bp were discarded. Finally, the filtered reads were merged using the BBMerge tool (Geneious Prime^®^ Version 2023.2.1). The rest of the analysis were conducted on the Easy Microbiome Analysis Platform (EasyMAP) (http://easymap.cgm.ntu.edu.tw/ (accessed on 25 May 2024)) following the recommendations made by Hung et al. [46]. The previously trimmed and merged sequences were clustered into Operational Taxonomic Units (OTUs) with a 97% of similarity. The Greengenes Database was used for taxonomic classification through the q2-feature-classifier plugin following the default settings [46]. The phylogenetic tree construction was performed using the q2-alignment and q2-phylogeny plugins.

Microbial alpha diversity metrics, including observed OTUs, Faith’s Phylogenetic Diversity (PD), Pielou Evenness, and Shannon index, were computed utilizing the q2-diversity plugin, to comprehensively assess microbial community richness and evenness. Additionally, rarefaction curves were generated to visualize the sampling depth and adequacy of sequencing coverage. The beta diversity was evaluated with a Principal coordinate analysis (PCoA) based on unweighted-UniFrac distance and Bray–Curtis dissimilarities. The statistical significance of the differences was assessed using permutational multivariate analysis of variance (PERMANOVA) with an alpha value of 0.05. Subsequently, for the discovery of potential biomarkers (taxonomic differential abundance), a linear discriminant analysis (LDA) effect size (LEfSe) was employed based on the KEGG database. The LEfSe algorithm was executed with the parameters determined by EasyMAP, including an alpha value of 0.05 for the Kruskal–Wallis and Wilcoxon rank-sum tests and a default threshold on the absolute value of the logarithmic (base 10) LDA score of 2. Finally, the microbiome functional profiling prediction was performed with the PICRUSt plugin, which used the mapped Greengenes ID to identify the corresponding functions in the KEGG database. The results were visualized with LEfSe.

The plots of taxonomy abundance, detected taxa, and alpha diversity indices were generated employing the software SigmaPlot version 14.5. Additionally, a statistical analysis was conducted to compare the number of bacterial taxa (in all levels) detected by each region and to ascertain the impact of hypervariable regions on alpha diversity metrics, using the same software. The normality and homogeneity of variance were assessed with the tests of Shapiro–Wilk and Brown–Forsythe, respectively. Data that passed these tests were evaluated with One-Way ANOVA, and for those that did not, the Kruskal–Wallis test was employed followed by post hoc Tukey’s test. In all cases, a significance level of *p* < 0.05 was used. All data presented in the results are expressed as the means ± standard deviation.

## 3. Results

### 3.1. Taxonomic Composition and Relative Abundance

A total of 1,524,672 filtered (Q > 30) sequences were obtained from 40 samples with an average of 38,117 ± 17,328 sequences per sample corresponding to a total of 4386 OTUs. The average taxonomic composition of the samples varied across hypervariable regions at the phylum, order, and genus levels (Figure 2). At the phylum level, the bacterial microbiota exhibited distinct dominance patterns across different hypervariable regions.

Notably, Firmicutes emerged as the predominant phylum, constituting 31% of the microbiota when the V3–V4 region was utilized. Conversely, Proteobacteria displayed higher abundance in the remaining regions. Also, the taxonomic classification efficiency varied across regions, with the V1–V2 region successfully assigning most sequences to at least the phylum level. On the contrary, the other regions ranked from most to least effective (V5–V7, V2–V3, V3–V4) yielded varying proportions of unclassified bacterial sequences. In terms of orders, distinct variations in taxonomic abundance were also observed among the regions. Clearly, Actinomycetales and Enterobacteriales emerged as the most prevalent taxa in the V1–V2, V2–V3, and V5–V7 regions (Figure 2b), comprising approximately 23% to 25% and 19% to 17% of the microbial community, respectively. Although, the V3–V4 region exhibited a contrasting profile, with Lactobacillales dominating (23%), followed by unclassified bacterial order (16%) and Bacteroidales (15%). Furthermore, the genus-level analysis revealed considerable variability compared to the phylum and order levels. Interestingly, while at least three regions shared the same most abundant taxon at higher taxonomic levels, the genus-level composition exhibited greater heterogeneity (Figure 2c). Predominant genera included *Propionibacterium* in V1–V2 (16%), *Escherichia* in V2–V3 (18%), *Streptococcus* in V3–V4 (21%), and *Enterobacteriaceae* in V5–V7 (16%).

Moreover, certain taxa exhibited region-specific identification at various taxonomic levels. For instance, the phylum Planctomycetes was exclusively detected by the V5–V7 hypervariable region, while the Cyanobacteria order (classified as “others” in Figure 2b) was uniquely associated with the V2–V3 region. At the genus level, the V3–V4 region was singular in its ability to distinguish the *Prevotella* genus, while *Haemophilus* was detected by both the V3–V4 and V5–V7 regions.

### 3.2. Number of Bacterial Taxa Detected by Each Hypervariable Region

The number of bacterial taxa detected in each taxonomic level were different among the 16S rRNA gene hypervariable regions (Figure 3).

A total of 9 phyla, 20 classes, 39 orders, 79 families, 108 genera, and 57 tentative species were captured among all regions. In general, the V3–V4 region identified the highest number of taxa at all levels except for the species level. In particular, the V3–V4 region revealed the detection of an average of 6 ± 0.1 phyla, significantly differing from the V1–V2 region (*p* = 0.002). Moreover, the V3–V4 region exhibited the identification of 11 ± 0.21 classes, representing a statistically higher abundance compared to other regions (*p* < 0.05). At the order level, the V2–V3 region demonstrated the lowest taxonomic diversity, encompassing an average of 13 ± 1.38 taxa, with significant differences observed compared to the V3–V4 region, which captured the highest number of orders (17 ± 0.44) (*p* < 0.05). Furthermore, the V3–V4 region outperformed other regions in capturing taxa at the family and genus levels, identifying 26 families and genera. This was followed by the V1–V2 region (22 families, 25 genera), V5–V7 region (22 families, 21 genera), and V2–V3 region (21 families, 21 genera). Notably, no statistically significant differences were detected among the regions (*p* > 0.05), indicating comparable performance in capturing taxa at this taxonomic level. Finally, the V1–V2 region found the highest number of species (11 ± 1.67) compared with the rest of the regions (V2–V3, V3–V4, and V5–V7 in order the major to minor), even so they were statistically similar (*p* = 0.11).

### 3.3. Microbial Alpha and Beta Diversity Metrics

The bacterial richness and alpha diversity indices varied between all regions (Figure 4). The V3–V4 region showed the highest richness with 425 ± 50.95 observed OTUs, followed by V1–V2 (296 ± 103.79), and both regions were statistically different (*p* < 0.05) from each other and the rest of the regions. The lowest number of OTUs were presented by V2–V3 (95 ± 44.82) and V5–V7 (138 ± 50.95) regions, with no significant differences between them (*p* > 0.05). The Shannon diversity index was significantly different (*p* < 0.05) in almost all amplified regions, with the highest diversity observed in the V3–V4 region (8.13 ± 0.24), followed by V1–V2, V5–V7, and V2–V3 (Figure 4b).

On the other hand, Faith’s PD was surprisingly higher in the V2–V3 (30 ± 9.89) and V5–V7 (16 ± 5.64) regions, which had previously presented less diversity when other indices were considered (Figure 4c). Statistically significant differences were observed in Faith’s PD among the hypervariable regions. Specifically, the V1–V2 region exhibited the lowest PD value at 7 ± 1.23, significantly differing from the PD value of the V2–V3 region (*p* < 0.001). Besides that, the PD value of the V3–V4 region was found to be similar to that of the V5–V7 region (*p* = 0.68). The bacterial communities of V3–V4 and V1–V2 regions were significantly (*p* < 0.001) more evenly distributed (Pielou’s evenness of 0.93 ± 0.01 and 0.92 ± 0.01, respectively) than V2–V3 region (Figure 4d). Also, this last one was statistically similar (*p* = 0.31) to the V5–V7 region. Figure 5 shows the results of the compositional dissimilarity analysis between the microbial communities (beta diversity) amplified with different regions. The PCoA plots based on Bray–Curtis dissimilarity (Figure 5a) and unweighted-UniFrac distances (Figure 5b), clearly illustrates a clustering of the bacterial communities into four different groups corresponding to each hypervariable region. Indicating a strong effect of the regions on the bacterial composition with highly significant differences (PERMANOVA, *p* = 0.001 for both indices) among the clusters.

### 3.4. Microbiome Differential Abundance and Function Prediction

The analysis of microbiome differential abundance carried out with LEfSe is shown in Figure 6, where the bar chart represents the distribution of differentially abundant taxa (at different levels) discriminated by each hypervariable region. A total of 39 high-dimensional biomarkers were discovered among all regions. Most of these were determined by the V1–V2 region (21), where the most significantly overrepresented taxa belong to the family Enterobacteriaceae and the genera *Propionibacterium* and *Staphylococcus* with a high LDA score > 8, highlighting the potential importance of these taxa as biomarkers of this region. Instead, the V3–V4 region was mostly associated with the Streptococcaceae and Pasteurellaceae families, the Lactobacillales order, and the Firmicutes phylum (LDA score > 6). A smaller number of differentially expressed taxa were identified by the rest of the regions. The taxa Listeriaceae (family) and *Escherichia* (genus) were discriminated by the V2–V3 region and the Comamonadaceae (family), Chloroplast (class) and *Gemella* (genus) by the V5–V7 region (Figure 6).

The microbiome function prediction identified with the integration of LEfSe and PICRUSt (based on the KEGG pathway second-level functional predictions) revealed different metabolic pathways in each hypervariable region (Figure 7). The microbial communities amplified with the V2–V3 region yielded the highest number of routes (10) and were mainly enriched with functions such as cell motility, xenobiotic biodegradation and metabolism, signal transduction, lipid metabolism, and energy metabolism (LDA score > 3). Conversely, the metabolic pathways more significantly abundant in the V3–V4 region were associated with replication and repair, translation, nucleotide metabolism, and glycan biosynthesis and metabolism (LDA score > 3). A smaller number of significant microbe functions were exhibited by the V1–V2 (4) and V5–V7 (5) regions. Amino acid metabolism and metabolism of cofactors and vitamins were the major metabolic pathways associated with the V5–V7 region (LDA score > 3). Meanwhile, the V1–V2 region suggested a higher potential for the carbohydrate metabolism and cellular processes and signaling (Figure 7).

## 4. Discussion

This study conducted a comprehensive analysis of the effects of four combinations of hypervariable regions (V1–V2, V2–V3, V3–V4, and V5–V7) from the 16S rRNA gene, revealing significant differences in the gut microbial diversity and composition of totoaba detected by each region. The significance of this study lies in determining which regions of the 16S rRNA gene provide the most taxonomic information [3]. A wide range of studies in the field of human health have focused on the resolution of different hypervariable regions to estimate the diversity of microbial communities in various sample types, such as vaginal specimens [16], breast tissue [25], urinary tract samples [47], infected tissue [23], and sputum samples [24]. However, few studies have addressed this in marine fish. The closest study in relationship to ours is by Klemetsen and collaborators, who compared the microbial profile of the skin mucus and the bulk intestinal content of Atlantic salmon using the V3–V4, V5–V6 regions, and the full-length 16S rRNA gene [48]. To our knowledge, this is the first study to examine the effect of 16S rRNA hypervariable region selection on the bacterial profile of the totoaba intestine, contributing to the expansion of knowledge in this area. This is crucial because the performance of hypervariable regions varies across different environments and organisms, as well as within the same organisms [17,42,49].

The totoaba gut microbial community composition identified by each region was principally dominated by Proteobacteria and Firmicutes phyla, followed by a lower proportion of Actinobacteria and Bacteroidetes. Other studies have shown similar results when analyzing the totoaba intestine [27,40,41], especially with Proteobacteria and Firmicutes phyla that could be related to the metabolic processes of proteins and carbohydrates, respectively [27]. Additionally, these phyla have been identified as the primary components of the totoaba gut core microbiota due to their consistent presence across various studies [35]. The core microbiota, also known as autochthonous microbiota, refers to microbial communities that remain stable over time and under different conditions because they are adhered to the intestinal epithelium [50]. However, in our study, the relative abundance of these phyla notably differed, particularly in the V3–V4 region. Additionally, this region detected the highest number of bacterial taxa across almost all taxonomic levels (Figure 3) and exhibited significantly higher alpha diversity indices (Figure 4) with the exception of Faith’s PD index.

Faith’s PD index measures biodiversity by accounting for the phylogenetic relationships among community members [51]. It is defined as the sum of the branch lengths in a phylogenetic tree that connects all the taxa present in a sample [51,52]. The low levels of this index observed in the V3–V4 region may be due to the taxa being phylogenetically closer. In contrast, the other alpha diversity indices used in this study do not consider the phylogeny of the taxa. The observed OTUs quantify species richness or the number of taxa [53], while the Shannon index combines richness and the relative abundance of taxa [54]. The Pielou evenness index considers the number of individuals or, in this case, sequences that belong to each taxon present in a sample [55]. Thus, the V3–V4 region emerged as the most optimal choice for amplifying microbial communities present in the totoaba intestine. Similar results have been obtained when comparing the performance of different regions in identifying bacterial microbiota from human stool samples and shrimp intestines, both studies showed that the most accurate region was V3–V4 [21,49]. Moreover, our principal coordinate analysis based on Bray–Curtis dissimilarity and unweighted-UniFrac distances (Figure 5) distinctly illustrates the impact of region selection, as evidenced by the formation of clearly distinct clusters corresponding to each studied region. This effect has been consistently reported in previous studies. For example, the V7–V9 and V1–V2 regions showed compositional differences in human sputum samples [24]. Additionally, beta diversity analyses of microbial communities in the Arctic Ocean revealed significant dissimilarity in the compositions of the microbiomes captured with the V3–V4 and V4–V5 regions [42].

The favorable outcomes observed with the V3–V4 region in this study may be attributed to the structural characteristics of its sequences. Notably, this region encompasses the C4 (conserved) region, which is distinguished by the presence of the most conserved sequence within the entire 16S rRNA gene [56]. This sequence comprises ten consecutive nucleotides (TTAGATACCC) [14]. The high conservation level of this segment is due to its function in binding tRNAs and relating with the 23S rRNA during the translation process [17,57]. The V3–V4 region also encompasses the V3 region, which is one of the most variable regions, containing twenty-three highly variable nucleotides within its sequence, this variability allows it to distinguish 110 bacterial genera [3,14]. This combination enhances the identification of a wide range of bacteria communities. The highly conserved sequence ensures the targeting of conserved regions across multiple bacterial taxa employing universal primers [14,57], while the highly variable sequence allows for phylogenetic differentiation and detects finer taxonomic differences between communities, in conjunction, they can be utilized to elucidate subtle variations in microbial functioning [3,17].

Another important factor that could have contributed to the high resolution of the results obtained with the V3–V4 region was the use of primers containing four degenerate bases (Appendix A). This means that these positions can match with several possible bases in the DNA template [58]. The degenerate primers are widely used for 16S rRNA gene sequencing because they allow the amplification of similar sequences but with genetic variation present in different organisms [59]. The use of such primers may have increased the capacity of the V3–V4 region to detect higher diversity in the microbiome of the totoaba gut (Figure 4). Nevertheless, studies have shown that these primers exhibit alignment to human mitochondrial DNA in breast tumors samples, in contrast to primers targeting the V1–V2 region. Interestingly, the latter demonstrated a notable reduction of approximately 80% in reads aligning to the human genome [25]. The same primers were employed in this study to amplify the V1–V2 region (refer to Appendix A), showcasing superior taxonomic classification efficiency and, notably, the complete absence of unassigned sequences. Typically, such sequences originate from non-target gene amplification, often associated with host DNA [25]. Our findings underscore the efficacy of the V1–V2 region, particularly in fish samples containing host DNA, such as intestinal tissue.

Currently, deciding on the optimal hypervariable region of the 16S rRNA gene for microbiome studies remains a challenge [21,24,47]. Our study reveals that bacterial taxonomic profiles within the totoaba intestine exhibit variation across each assessed region. It remains challenging to ascertain which region most accurately captures the true taxonomic profile, primarily due to the absence of a defined microbial community within the totoaba intestine, as far as our knowledge extends. This limitation has been described in other studies, for example Heidrich et al. [47] determined that it is virtually impossible to fully infer the microbiota profile from male urinary samples employing 16S rRNA amplicon sequencing. Moreover, it has been shown that no single hypervariable region is able to distinguish among all bacteria [3]. One way to overcome this limitation is to sequence the full-length 16S rRNA gene, as covering all hypervariable regions may increase the accuracy and resolution of the analysis, providing more detailed genetic information and improving the identification of bacterial taxa [21,48]. This can be achieved using third-generation sequencing technologies, such as PacBio, which employ circular consensus sequencing (CCS) to improve the accuracy of sequencing. This approach generates long reads lengths (13.5 kb) with high accuracy [60]. Furthermore, Klemetsen et al. [48] compared the performance of full-length 16S rRNA gene sequences obtained by PacBio sequencing against partial sequences spanning the V3–V4 and V5–V6 regions, which were extracted in silico from the full-length 16S rRNA gene sequence dataset. Their results showed that the V3–V4 region had the highest proportion of assigned sequences at the family and genus levels, while the full-length dataset had the highest proportion of assignments at the species level. Nevertheless, third-generation sequencing technologies have disadvantages, such as higher costs, relatively high error rates, and less standardized protocols and analysis pipelines [21,61,62].

The four combinations of hypervariable regions from the 16S rRNA gene assessed in this study showed differentially abundant taxa at various taxonomic levels (Figure 6). These taxa can be used as specific high-dimensional biomarkers for each region and may explain the differences observed between the hypervariable regions [63]. Moreover, these biomarkers influence the functionality of the microbiome detected by each region. The integration of LEfSe and PICRUSt utilizes the relative abundance of taxa within the community, based on the reference genome for each taxon present [64]. Consequently, differentially expressed taxa resulted in distinct metabolic pathways detected by each hypervariable region (Figure 7).

This study demonstrates the impact of selecting the hypervariable region of the 16S rRNA gene on the analysis of the totoaba fish microbiome. It also provides a foundation for future research by offering a methodological advancement that enhances the accuracy of fish microbiota analysis, particularly in species of ecological, aquaculture, and commercial importance, such as the totoaba. Since a balanced microbiota is essential for fish development and provides protection against pathogenic microbes, which are a significant cause of mortality in fish production systems [65]. Furthermore, our study not only identifies the most efficient hypervariable region (V3–V4) for analyzing the totoaba microbiota but also provides a taxonomic profile for each region tested with a differential abundance analysis and microbiome function prediction. This will enable future researchers to determine whether their microbiome studies are influenced by the selected region or by other factors under investigation. Since the microbiome function prediction can serve as an indicator of the host’s health, such as the presence of a pathogen [66,67], understanding these influences is crucial.

Additionally, we identify unique taxa that were only detected by specific regions, which simplifies future research when targeting specific taxa. However, as with all studies utilizing 16S rRNA amplicon sequencing, the cost of sequencing was a limitation. Although a larger sample size, additional replicates, and the inclusion of more 16S rRNA hypervariable regions (such as V7–V8 and V8–V9) would have provided greater depth, the available data still allowed for a comprehensive analysis. Another limitation of this study is that we only analyzed juvenile fish. Including various life stages of totoaba could have provided additional insights into the effects of region selection on the bacterial microbiota throughout the organism’s life cycle, especially because the gut microbiota composition changes at each life stage [68]. However, due to the endangered status of this species, organism availability was understandably limited. Despite these constraints, our robust analysis and results demonstrate that region selection significantly alters the bacterial taxonomic profile of *T. macdonaldi*. These findings remain credible and valuable, even within the scope of this study’s inherent limitations. Moreover, an interesting way to enrich future microbiome studies is by incorporating other techniques, such as transmission electron microscopy (TEM), which allows for the ultrastructural visualization of microbial communities. This is particularly useful when the focus is on specific taxon or species [69,70]. Also, implementing fluorescent in situ hybridization (FISH) can be valuable for studying microbial composition and spatial structure [31,71].

## 5. Conclusions

The V3–V4 region demonstrated the highest taxonomic certainty, establishing it as the optimal choice for identifying the microbiome of the totoaba intestine. Furthermore, the V1–V2 region displayed the highest efficiency in taxonomic classification. This study underscores the critical importance of 16S rRNA region selection in fish microbiota analysis. Additionally, the distinct taxonomic profiles detected for each region offer a valuable foundation for future research, enabling targeted investigations into specific bacterial taxa. By facilitating more accurate microbial identification, our study contributes to a deeper understanding of fish microbiomes, which is essential for ecological studies and effective disease management strategies.

## Figures and Tables

**Figure 1 microorganisms-12-02119-f001:**
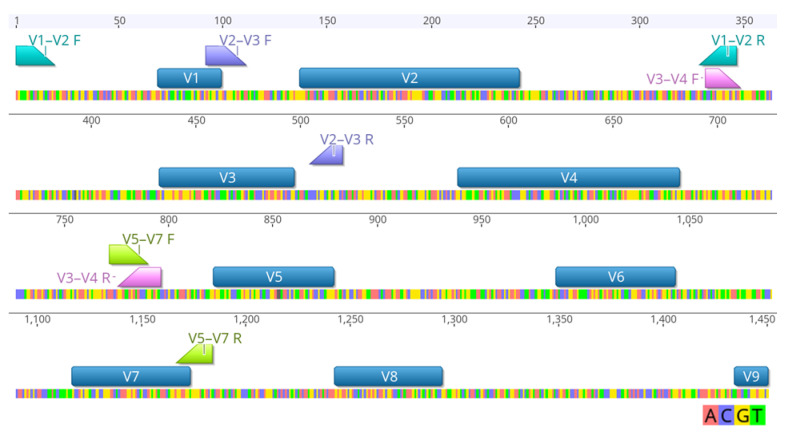
Representation of the 16S rRNA gene hypervariable regions. The positions of the utilized primers (forward and reverse) for the amplification of four hypervariable regions are shown. The start and finish of the V1–V2 (aqua), V2–V3 (purple), V3–V4 (pink), and V5–V7 (green) regions are highlighted. The positioning is based on the *E. coli* strain U 5/41 16S rRNA gene.

**Figure 2 microorganisms-12-02119-f002:**
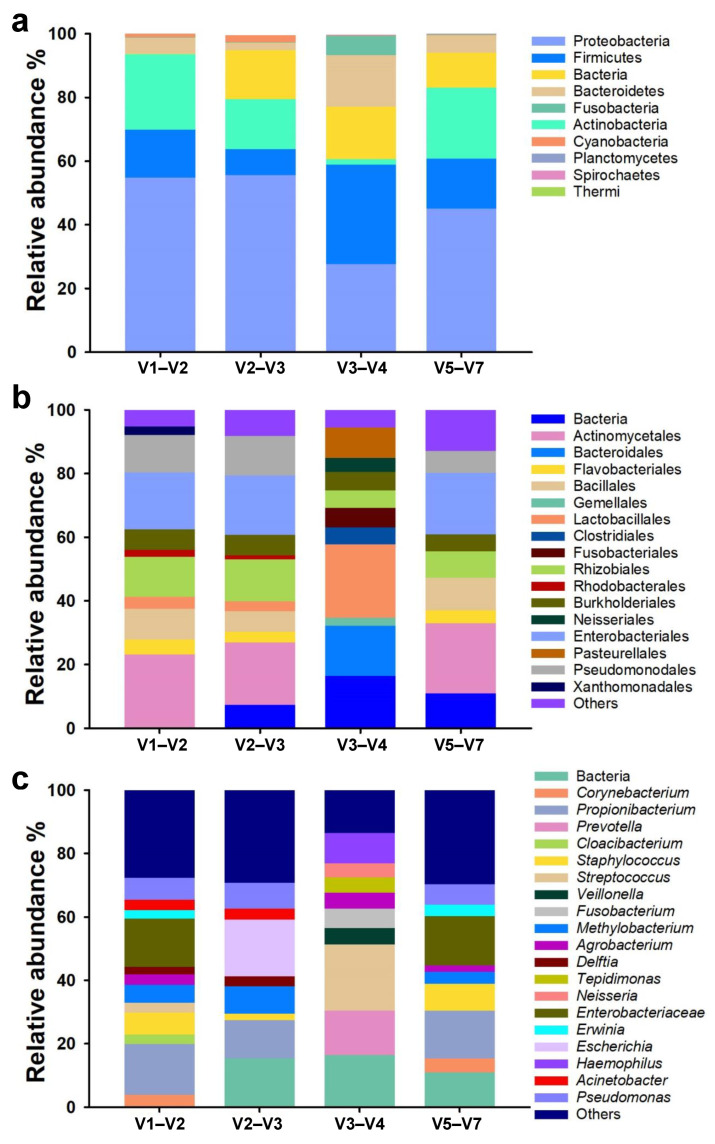
Taxonomic compositions and relative abundance of the microbial communities present in the intestine of *T. macdonaldi*. Samples were grouped and averaged by the hypervariable region at (**a**) the phyla, (**b**) order and (**c**) genus levels. The taxa averaged at the order and genus levels with relative abundance below 2% were classified as others.

**Figure 3 microorganisms-12-02119-f003:**
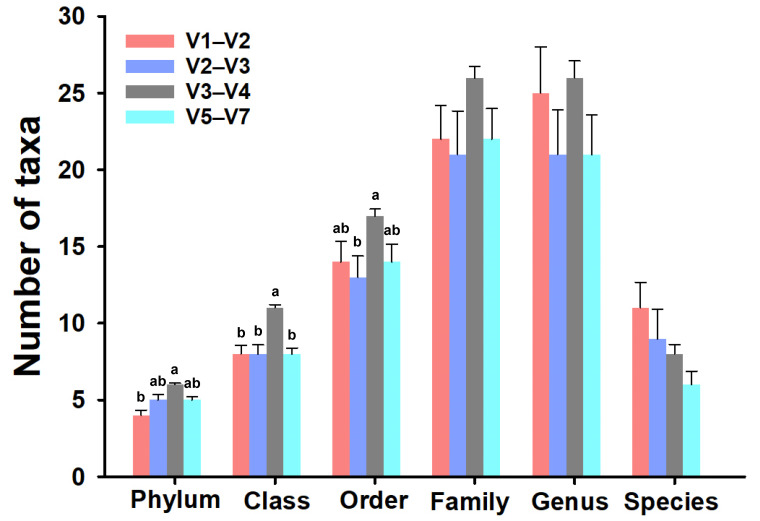
Number of taxa detected by each hypervariable region. The bars represent the average number of taxa detected in all samples by each region at different taxonomy levels. Error bars represent the standard error for each group (*n* = 10). Different letters indicate significant differences between groups (*p* < 0.05).

**Figure 4 microorganisms-12-02119-f004:**
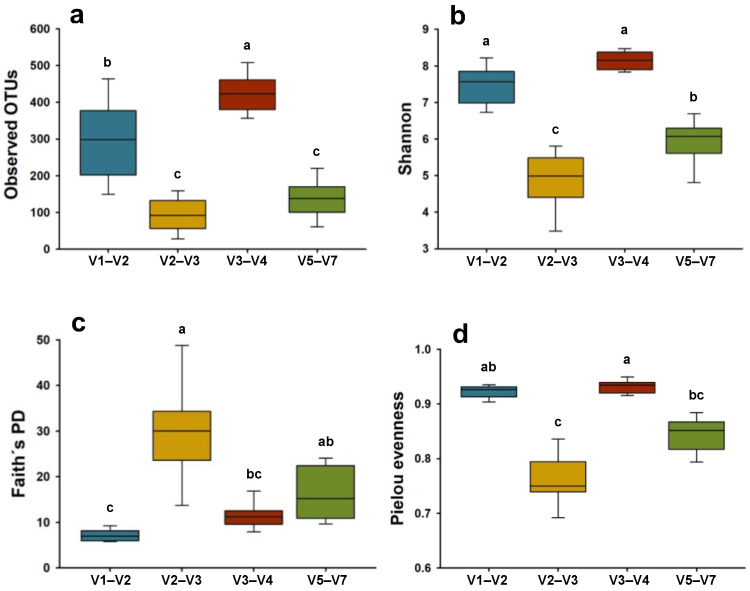
Alpha diversity indices of microbiomes amplified with different hypervariable regions from the intestine of *T. macdonaldi* fish. The (**a**) observed OTUs, (**b**) Shannon index, (**c**) Faith’s PD, and (**d**) Pielou evenness were calculated for the average of all samples (*n* = 10) by each hypervariable region. Different letters indicate significant differences between groups (*p* < 0.05).

**Figure 5 microorganisms-12-02119-f005:**
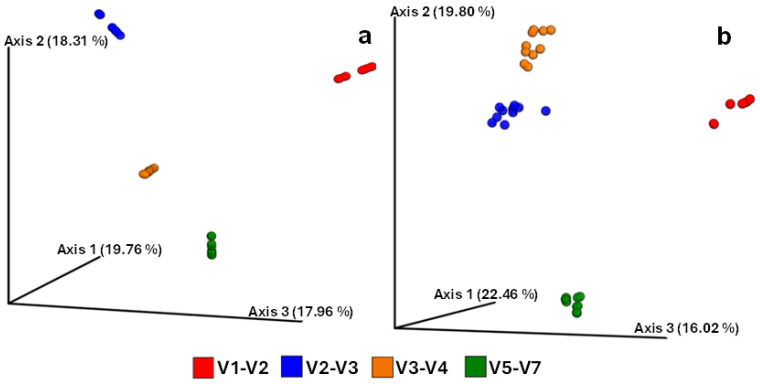
Beta diversity indices of microbiomes amplified with different hypervariable regions from the intestine of *T. macdonaldi* fish. The plot shows a PCoA analysis based on (**a**) Bray–Curtis dissimilarity and (**b**) unweighted-UniFrac distance of bacterial communities composition.

**Figure 6 microorganisms-12-02119-f006:**
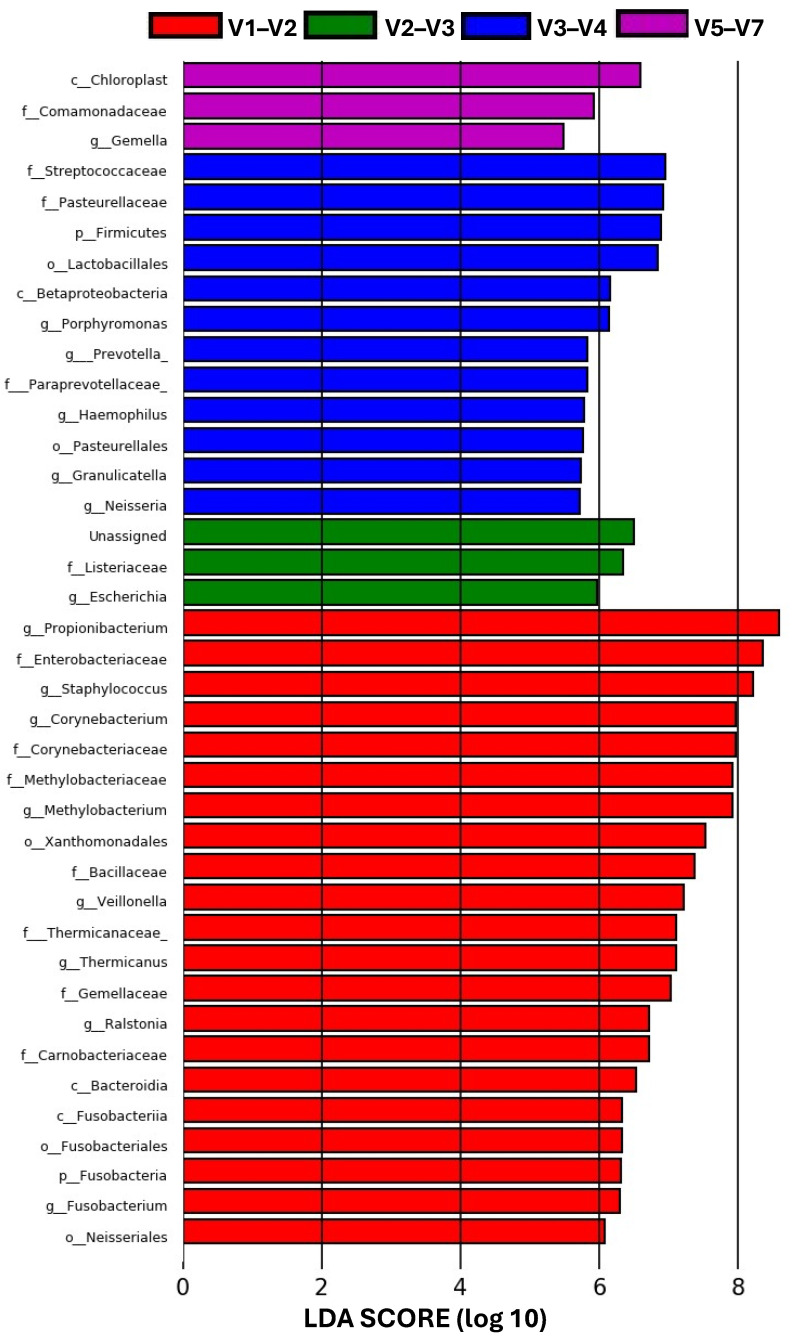
Differentially abundant taxa discriminated by different hypervariable regions from the intestine of *T. macdonaldi* fish. The bar chart displays the results of the LEfSe analysis, with bars representing statistically significant differences in bacterial abundance at different taxonomic levels. Only taxa meeting an LDA significance threshold of >2 are shown.

**Figure 7 microorganisms-12-02119-f007:**
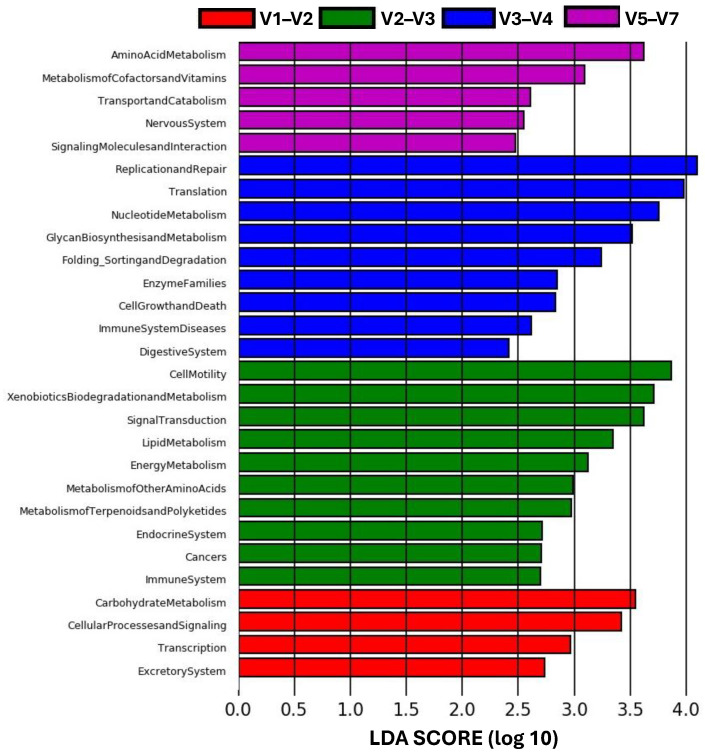
Microbiome function prediction detected by different hypervariable regions from the intestine of *T. macdonaldi* fish. The bars represent the significantly different metabolic pathways based on KEGG level 2. Only pathways with an LDA significant threshold of >2 are shown.

## Data Availability

All raw data are available in the NCBI repository at https://www.ncbi.nlm.nih.gov/sra/PRJNA1147430 (accessed on 29 August 2024) under accession number PRJNA1147430.

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
