# Peer review of "Effect of the 16S rRNA Gene Hypervariable Region on the Microbiome Taxonomic Profile and Diversity in the Endangered Fish Totoaba macdonaldi"

_microorganisms, 2024, doi:10.3390/microorganisms12112119_

Round 1
Reviewer 1 Report
Comments and Suggestions for Authors
Dear authors,
Congratulations for your work results.
Please follow the below suggestions:
Not all the not-original statements have citations.
In the general the citations and final references should be enriched.
Try to explain why you consider that 5 only cultured and only juvenile fish are enough to stress the paper conclusions OR explain specifically and accept the limitations of your study/results.
Comparison between fish-human situation is relevant? Why?
The study results limitations should be better described in the context of the paper.
Also the importance of the results should be highlighted and developed.
All the best and success!
Reviewer
Author Response
Comments 1: Not all the not-original statements have citations |
Response 1: Thank you for bringing this to our attention. We have carefully reviewed the entire document and added citations to all statements that were not originally referenced as needed. While we are unsure which specific statement you were referring to, we hope that our revisions adequately address your concerns. Line 34: we added [1–3] references (including one new reference). Line 35: we added “and has been employed to identify bacterial populations from various hosts, including algae [4], mollusks [5], crustaceans [6], and fish [7]” including three new references. Line 36: we added the reference number [8]. Line 37: we added the reference number [9]. Line 39: we added the references [10,11]. Line 40: we added the reference [12] Line 41: we added the references [13,14]. Line 42: we added the reference [15]. Line 43: we added the references [2,14]. Line 46: we added the references [9,12]. Line 47: we added the references [3,14]. Line 51: we added the references [16–18]. Line 52: we added the references [19,20]. Line 64: we added the references [7,21–23]. Line 94: we added the reference [24]. Line 360: we added the reference [25]. Line 383: we added the reference [26]. Line 391: we added the reference [14,27] |
Comments 2: In the general the citations and final references should be enriched Response 2: We appreciate your comment and have incorporated additional references throughout the document, particularly in the introduction section where we felt they were most necessary. The updated list of references is provided below: |
1. Srinivasan, R.; Karaoz, U.; Volegova, M.; MacKichan, J.; Kato-Maeda, M.; Miller, S.; Nadarajan, R.; Brodie, E.L.; Lynch, S. V. Use of 16S RRNA Gene for Identification of a Broad Range of Clinically Relevant Bacterial Pathogens. PLoS One 2015, 10, doi:10.1371/journal.pone.0117617. 4. Kim, K.H.; Kim, J.M.; Baek, J.H.; Jeong, S.E.; Kim, H.; Yoon, H.S.; Jeon, C.O. Metabolic Relationships between Marine Red Algae and Algae-Associated Bacteria. Mar Life Sci Technol 2024, 6, 298–314, doi:10.1007/s42995-024-00227-z. 5. Scanes, E.; Parker, L.M.; Seymour, J.R.; Siboni, N.; Dove, M.C.; O’Connor, W.A.; Ross, P.M. Microbiomes of an Oyster Are Shaped by Metabolism and Environment. Sci Rep 2021, 11, doi:10.1038/s41598-021-00590-2. 6. Li, M.; Ghonimy, A.; Chen, D.Q.; Li, J.T.; He, Y.Y.; López Greco, L.S.; Dyzenchauz, F.; Chang, Z.Q. Profile of the Gut Microbiota of Pacific White Shrimp under Industrial Indoor Farming System. Appl Microbiol Biotechnol 2024, 108, doi:10.1007/s00253-024-13046-0. 7. Soh, M.; Tay, Y.C.; Lee, C.S.; Low, A.; Orban, L.; Jaafar, Z.; Seedorf, H. The Intestinal Digesta Microbiota of Tropical Marine Fish Is Largely Uncultured and Distinct from Surrounding Water Microbiota. NPJ Biofilms Microbiomes 2024, 10, doi:10.1038/s41522-024-00484-x. 8. Woese, C.R.; Kandlert, O.; Wheelis, M.L. Towards a Natural System of Organisms: Proposal for the Domains Archaea, Bacteria, and Eucarya. Proc. Natl. Acad. Sci. 1990, 87, 4576-4579. 9. Wang, Y.; Qian, P.Y. Conservative Fragments in Bacterial 16S RRNA Genes and Primer Design for 16S Ribosomal DNA Amplicons in Metagenomic Studies. PLoS One 2009, 4, doi:10.1371/journal.pone.0007401. 12. Clarridge, J.E. Impact of 16S RRNA Gene Sequence Analysis for Identification of Bacteria on Clinical Microbiology and Infectious Diseases. Clin Microbiol Rev 2004, 17, 840–862. 15. Patel, J. 16S RRNA Gene Sequencing for Bacterial Pathogen Identification in the Clinical Laboratory. Molecular Diagnosis 2001, 6, 313–321, doi:10.1054/modi.2001.29158. 17. Yang, B.; Wang, Y.; Qian, P.Y. Sensitivity and Correlation of Hypervariable Regions in 16S RRNA Genes in Phylogenetic Analysis. BMC Bioinformatics 2016, 17, doi:10.1186/s12859-016-0992-y. 18. Bharti, R.; Grimm, D.G. Current challenges and best-practice protocols for microbiome analysis. Brief. Bioinform. 2021, 22 (1), 178-193. 23. Jang, W.J.; Kim, S.K.; Lee, S.J.; Kim, H.; Ryu, Y.W.; Shin, M.G.; Lee, J.M.; Lee, K.B.; Lee, E.W. Effect of Bacillus Sp. Supplementation Diet on Survival Rate and Microbiota Composition in Artificially Produced Eel Larvae (Anguilla Japonica). Front Microbiol 2022, 13, doi:10.3389/fmicb.2022.891070. 26. Ram, J.L.; Karim, A.S.; Sendler, E.D.; Kato, I. Strategy for Microbiome Analysis Using 16S RRNA Gene Sequence Analysis on the Illumina Sequencing Platform. Syst Biol Reprod Med 2011, 57, 162–170, doi:10.3109/19396368.2011.555598. 30. Cornuault, J.K.; Byatt, G.; Paquet, M.E.; De Koninck, P.; Moineau, S. Zebrafish: A Big Fish in the Study of the Gut Microbiota. Curr Opin Biotechnol 2022, 73, 308–313. 31. Torge, D.; Bernardi, S.; Ciciarelli, G.; Macchiarelli, G.; Bianchi, S. Dedicated Protocol for Ultrastructural Analysis of Farmed Rainbow Trout (Oncorhynchus Mykiss) Tissues with Red Mark Syndrome: The Skin—Part One. Methods Protoc 2024, 7, doi:10.3390/mps7030037. 32. Watteau, F.; Villemin, G. Soil Microstructures Examined through Transmission Electron Microscopy Reveal Soil-Microorganisms Interactions. Front Environ Sci 2018, 6, doi:10.3389/fenvs.2018.00106. 34. McCallum, G.; Tropini, C. The Gut Microbiota and Its Biogeography. Nat Rev Microbiol 2024, 22, 105–118. 36. Vargas-Albores, F.; Martínez-Córdova, L.R.; Hernández-Mendoza, A.; Cicala, F.; Lago-Lestón, A.; Martínez-Porchas, M. Therapeutic Modulation of Fish Gut Microbiota, a Feasible Strategy for Aquaculture? Aquaculture 2021, 544. 37. López-Carvallo, J.A.; Cruz-Flores, R.; Dhar, A.K. The Emerging Pathogen Enterocytozoon Hepatopenaei Drives a Degenerative Cyclic Pattern in the Hepatopancreas Microbiome of the Shrimp (Penaeus Vannamei). Sci Rep 2022, 12, 14766, doi:10.1038/s41598-022-19127-2.
|
Comments 3: Try to explain why you consider that 5 only cultured and only juvenile fish are enough to stress the paper conclusions OR explain specifically and accept the limitations of your study/results. |
Response 3: We understand your concern regarding the sample size, which was limited to five fish due to the costs associated with sequencing. However, this sample size allowed us to conduct duplicate sequencing for each sample, ensuring the robustness and reliability of the data, while also enabling us to study a greater number of regions (4). Our decision was further guided by the sample sizes used in similar studies [28,29]. Nonetheless, we acknowledge this limitation and have included a detailed explanation in the paragraph below: Line 458-475: “However, as with all studies utilizing 16S rRNA amplicon sequencing, the cost of sequencing was a limitation. Although a larger sample size, additional replicates, and the inclusion of more 16S rRNA hypervariable regions (such as V7-V8 and V8-V9) would have provided greater depth, the available data still allowed for a comprehensive analysis. Another limitation of this study is that we only analyzed juvenile fish. Including various life stages of totoaba could have provided additional insights into the effects of region selection on the bacterial microbiota throughout the organism's life cycle, especially because the gut microbiota composition changes at each life stage [28]. However, due to the endangered status of this species, organism availability was understandably limited. Despite these constraints, our robust analysis and results demonstrate that region selection significantly alters the bacterial taxonomic profile of T. macdonaldi. These findings remain credible and valuable, even within the scope of the study's inherent limitations. Moreover, an interesting way to enrich future microbiome studies is by incorporating other techniques, such as transmission electron microscopy (TEM), which allows for the ultrastructural visualization of microbial communities. This is particularly useful when the focus is on specific taxon or species [29,30]. Also, implementing fluorescent in situ hybridization (FISH) can be valuable for studying microbial composition and spatial structure [31,32]”. R Comments 4: Comparison between fish-human situation is relevant? Why? Response 4: Comparisons between humans and fish are common in aquaculture research, especially when studying new topics where existing studies are limited. In such cases, we often rely on insights from more established fields, such as human health. Similar comparisons have been made in other studies [11,24,33]. Furthermore, our study demonstrates that drawing on human studies can accelerate knowledge acquisition and scientific progress in aquaculture by providing a more efficient framework for research. Specifically, we refer to the selection of the V1-V2 region that was based on a human study carried out by Walker et al. [34], as no prior studies have examined this region in fish. Our study yielded favorable results as you have been described on lines 403-409. 11. Reigel, A.M.; Owens, S.M.; Hellberg, M.E. Reducing Host DNA Contamination in 16S RRNA Gene Surveys of Anthozoan Microbiomes Using PNA Clamps. Coral Reefs 2020, 39, 1817–1827, doi:10.1007/s00338-020-02006-5. Lópe2 24. Aladid, R.; Fernández-Barat, L.; Alcaraz-Serrano, V.; Bueno-Freire, L.; Vázquez, N.; Pastor-Ibáñez, R.; Palomeque, A.; Oscanoa, P.; Torres, A. Determining the Most Accurate 16S RRNA Hypervariable Region for Taxonomic Identification from Respiratory Samples. Sci Rep 2023, 13, doi:10.1038/s41598-023-30764-z. 33. Zhong, X.; Li, J.; Lu, F.; Zhang, J.; Guo, L. Application of Zebrafish in the Study of the Gut Microbiome. Animal Model Exp Med 2022, 5, 323–336. 34 34. Walker, S.P.; Barrett, M.; Hogan, G.; Flores Bueso, Y.; Claesson, M.J.; Tangney, M. Non-Specific Amplification of Human DNA Is a Major Challenge for 16S RRNA Gene Sequence Analysis. Sci Rep 2020, 10, doi:10.1038/s41598-020-73403-7.
Comments 5: The study results limitations should be better described in the context of the paper. Response 5: We agree with your comment, and the limitations have been described on lines 458-475. Additionally, you can find the paragraph in the response to your comment number 3.
Comments 6: Also the importance of the results should be highlighted and developed. Response 6: Thank you for pointing this out. We have added a paragraph to highlight the importance of our results, as shown below: Line 443-458: “This study demonstrates the impact of selecting the hypervariable region of the 16S rRNA gene on the analysis of the totoaba fish microbiome. It also provides a foundation for future research by offering a methodological advancement that enhances the accuracy of fish microbiota analysis, particularly in species of ecological, aquaculture, and commercial importance, such as the totoaba. Since a balanced microbiota is essential for fish development and provides protection against pathogenic microbes, which are a significant cause of mortality in fish production systems [35]. Furthermore, our study not only identifies the most efficient hypervariable region (V3-V4) for analyzing the totoaba microbiota but also provides a taxonomic profile for each region tested with a differential abundance analysis and microbiome function prediction. This will enable future researchers to determine whether their microbiome studies are influenced by the selected region or by other factors under investigation. Since the microbiome function prediction can serve as an indicator of the host's health, such as the presence of a pathogen [36], understanding these influences is crucial. Additionally, we identify unique taxa that were only detected by specific regions, which simplifies future research when targeting specific taxa.”
|
5. Additional clarifications |
All queries raised by the reviewers have been addressed.
|

Reviewer 2 Report
Comments and Suggestions for Authors
The aim of this manuscript is to conduct a deep investigation to identify the most suitable 16S rRNA region for bacterial microbial analysis in juveniles of T. macdonaldi.
There are some suggestions necessary to make the article complete and fully readable. For these reasons, the manuscript requires major changes.
Please find below an enumerated list of comments on my review of the manuscript:
MINOR POINTS:
There is no list of the abbreviations, mentioned in this manuscript. If possible, please, provide it.
MAJOR POINTS:
INTRODUCTION:
LINE 69: Totoaba macdonaldi is an endemic, vulnerable, carnivorous fish of the Gulf of California that is currently being cultivated in northwestern Mexico for commercial and conservation purposes. The challenge for aquaculture of Totoaba is finding a diet that meets the protein requirements at an acceptable price and does not compromise its overall performance (see, for reference: https://doi.org/10.1016/j.aqrep.2023.101654).
CONCLUSIONS:
LINE 447: Furthermore, the development of microscopical techniques, such as transmission electron microscopy (TEM), revolutionized the morphological sciences, progressively providing new levels of magnification and resolution for exploring biological and non-biological samples. For these reasons, these techniques play a crucial role in discovering and describing the ultrastructural profile of bacteria and viruses (see, for reference: https://doi.org/10.3390/mps7030037). Are there some studies, which also performed morphological analyses, by means of microscopic techniques, in order to describe the microbiological profile in T. Macdonaldi? The manuscript may benefit from mention the contribute of microscopic analyses to the clarification of the microbiological profile, maybe analyzing the morphology of ribosomes: this could be a further future perspective of this study, in order to provide a morpho-functional overview of this issue.
As regards the section of methods, there is a specific and detailed explanation for the methods used in this study: this is particularly significant, since the manuscript relies on a multitude of methodological and statistical analysis, to derive its conclusions. The methodology applied is overall correct, the results are reliable and adequately discussed.
Finally, this manuscript also shows a basic structure, properly divided and looks like very informative on this topic. Furthermore, figures and tables are complete, organized in an organic manner and easy to read. However, major concerns of this manuscript are with the introductive and conclusive sections: for these reasons, I have major comments for these sections, for improvement before acceptance for publication. The article is accurate and provides relevant information on the topic and I have some major points to make, that may help to improve the quality of the current manuscript and maximize its scientific impact. I would accept this manuscript if the comments are addressed properly.
The main question of this research is to examine the most suitable 16S rRNA region for bacterial microbial analysis in juveniles of T. macdonaldi.
The originality and strengths of this manuscript are due to the organic and realistic description of the challenges, and specifically of the dynamics of fish microbiota through 16S rRNA amplicon sequencing. This comparative approach improves the quality and the scientific impact of this manuscript.
When compared to other published materials on this specific topic, this manuscript adds a significant contribute to this field of research: specifically, the topic chosen for this manuscript has been a hot topic, involving an endemic species of significant ecological and economic importance in Mexico. For these reasons, the authors of this manuscript have done significant work-up via searches of these database resources and the manuscript deals with the most recent scientific literature in this field: the authors do not limit to mention each study, but they discussed their results in the light of recent available scientific evidence.
Furthermore, the main questions of this manuscript were the following: to understand the intricate dynamics of fish microbiota through 16S rRNA amplicon sequencing. According to me, the authors fully answer to this first question, deeply revising the current scientific evidence on this topic, and providing complete illustrative schemes and figures, adherent to the topic.
The references included in this manuscript are appropriate and in line with the current scientific evidence on this topic. Anyway, as previously suggested, the manuscript will benefit from providing, in the introductive section of this manuscript, a brief description of Totoaba macdonaldi, as an endemic, vulnerable, carnivorous fish of the Gulf of California that is currently being cultivated in northwestern Mexico for commercial and conservation purposes. Finally, it will be useful to dedicate a brief space to the role of microscopic techniques in discovering and describing the ultrastructural profile of bacteria and viruses, as highlighted by recent evidence.

Author Response
Comments 1: There is no list of the abbreviations, mentioned in this manuscript. If possible, please, provide it. |
Response 1: Thank you for pointing this out. We have added a list of abbreviations in the lines 508-521.
|
Comments 2: LINE 69: Totoaba macdonaldi is an endemic, vulnerable, carnivorous fish of the Gulf of California that is currently being cultivated in northwestern Mexico for commercial and conservation purposes. The challenge for aquaculture of Totoaba is finding a diet that meets the protein requirements at an acceptable price and does not compromise its overall performance (see, for reference: https://doi.org/10.1016/j.aqrep.2023.101654). Response 2: We agree with your comment and have modified the paragraph according to your suggestion, including the additional information and reference. The final version of the text is shown below: Line 72-85: “The totoaba (Totoaba macdonaldi) is a vulnerable, carnivorous fish endemic to the Gulf of California and has become a keystone species in this region [37,38]. However, unregulated fishing and overexploitation have led to a dramatic decline in its wild population, resulting in its classification as an endangered species in Mexico (NOM-059-SEMARNAT-2010) [39] and a vulnerable species by the IUCN [40]. In recent years, aquaculture efforts have transformed totoaba cultivation into a significant economic activity, supporting the livelihoods of many fishermen throughout the Baja California peninsula [41]. Today, totoaba cultivation is well-established in the aquaculture industry of northwest Mexico, aimed at both enhancing wild stocks and commercial production [42]. However, a major challenge for totoaba aquaculture is developing a diet that meets its protein requirements at an adequate cost without compromising its overall performance [38].“ |
|
Comments 3: LINE 447: Furthermore, the development of microscopical techniques, such as transmission electron microscopy (TEM), revolutionized the morphological sciences, progressively providing new levels of magnification and resolution for exploring biological and non-biological samples. For these reasons, these techniques play a crucial role in discovering and describing the ultrastructural profile of bacteria and viruses (see, for reference: https://doi.org/10.3390/mps7030037). Are there some studies, which also performed morphological analyses, by means of microscopic techniques, in order to describe the microbiological profile in T. Macdonaldi? The manuscript may benefit from mention the contribute of microscopic analyses to the clarification of the microbiological profile, maybe analyzing the morphology of ribosomes: this could be a further future perspective of this study, in order to provide a morpho-functional overview of this issue. |
Response 3: We understand your comment. To our knowledge there are not studies describing microbiological profile of totoaba. Instead, we resalt the importance of the use of TEM in the lines 470-475.
Comments 4: As regards the section of methods, there is a specific and detailed explanation for the methods used in this study: this is particularly significant, since the manuscript relies on a multitude of methodological and statistical analysis, to derive its conclusions. The methodology applied is overall correct, the results are reliable and adequately discussed. Finally, this manuscript also shows a basic structure, properly divided and looks like very informative on this topic. Furthermore, figures and tables are complete, organized in an organic manner and easy to read. However, major concerns of this manuscript are with the introductive and conclusive sections: for these reasons, I have major comments for these sections, for improvement before acceptance for publication. The article is accurate and provides relevant information on the topic and I have some major points to make, that may help to improve the quality of the current manuscript and maximize its scientific impact. I would accept this manuscript if the comments are addressed properly. The main question of this research is to examine the most suitable 16S rRNA region for bacterial microbial analysis in juveniles of T. macdonaldi. Response 3: The authors thank the reviewer for the critical analysis of the manuscript. We have addressed the comment regarding the importance of adequate dietary protein in lines 72-85. Additionally, we have strengthened the discussion sections to highlight potential perspectives for studying the biogeography of the microbiota through microscopic observations (lines 470-475). Furthermore, changes to improve clarity and the inclusion of additional references have been made throughout the manuscript, as suggested. |
5. Additional clarifications |
All queries raised by the reviewers have been addressed.

Round 2
Reviewer 1 Report
Comments and Suggestions for Authors
Congratulations.
Success in your future studies, developing this paper valuable results.
Reviewer
Reviewer 2 Report
Comments and Suggestions for Authors
Congratulations to the authors, which have significantly improved the scientific impact and quality of the manuscript.